# An Easy-To-Use Survival Score Compared to Existing Tools for Older Patients with Cerebral Metastases from Colorectal Cancer

**DOI:** 10.3390/cancers12040833

**Published:** 2020-03-30

**Authors:** Dirk Rades, Trang Nguyen, Stefan Janssen, Steven E. Schild

**Affiliations:** 1Department of Radiation Oncology, University of Luebeck, 23570 Luebeck, Germany; trangnguyen.dv@googlemail.com (T.N.); st-janssen@gmx.net (S.J.); 2Medical Practice for Radiotherapy and Radiation Oncology, 30159 Hannover, Germany; 3Department of Radiation Oncology, Mayo Clinic, Scottsdale, AZ 85250, USA; sschild@mayo.edu

**Keywords:** colorectal cancer, older persons, cerebral metastases, radiation therapy, survival scores

## Abstract

An easy-to-use survival score was developed specifically for older patients with cerebral metastases from colorectal cancer, and was compared to existing tools regarding the accuracy of identifying patients who die in ≤6 months and those who survive for ≥6 months. The new score was built from 57 patients receiving whole-brain irradiation. It included three groups identified from 6-month survival rates based on two independent predictors (performance status and absence/presence of non-cerebral metastases), with 6-month survival rates of 0% (0 points), 26% (1 point), and 75% (2 points), respectively. This score was compared to diagnosis-specific scores, namely the diagnosis-specific graded prognostic assessment (DS-GPA), the Dziggel-Score and the WBRT-30-CRC (whole-brain radiotherapy with 30 Gy in 10 fractions for cerebral metastases from colorectal cancer) score and to a non-diagnosis-specific score for older persons (Evers-Score). Positive predictive values were 100% (new score), 87% (DS-GPA), 86% (Dziggel-Score), 91% (WBRT-30-CRC), and 100% (Evers-Score), respectively, for patients dying ≤6 months, and 75%, 33%, 75%, 60%, and 45%, respectively, for survivors ≥6 months. Of the five tools, the new score and the Evers-Score were most precise in identifying patients dying ≤6 months. The new score and the Dziggel-Scores were best at identifying patients surviving ≥6 months. When combining the results, the new score appeared preferable to the existing tools. The score appears not necessary for patients with additional liver metastases, since their 6-month survival rate was 0%.

## 1. Introduction

Colorectal cancer ranks as the third most common malignant disease worldwide, with an expected increase of the number of deaths until 2035 [1,2,3]. When compared to other cancer types, cerebral metastases occur less frequently, in only 1–2% of patients having colorectal cancers [4,5,6]. Many of these patients are treated with radiation therapy that, depending on the number and size of cerebral lesions, is administered locally as single-fraction radiosurgery and fractionated stereotactic radiation therapy, or as treatment of the entire brain (whole-brain irradiation) [7]. Both local irradiation and whole-brain irradiation can also be combined when appropriate. Local radiation therapy is generally limited to ≤ 5 cerebral lesions with diameters of ≤ 3–4 cm. Many other patients receive whole-brain irradiation. For this type of radiation, different dose fractionations exist, including one-week treatment with 20 Gy in five fractions, two-week treatment with 30 Gy in 10 fractions, or treatment with doses beyond 30 Gy lasting up to four weeks [7]. Each of these programs is not optimal for each patient. For example, patients with shorter expected survival times should be considered for 20 Gy in five fractions, to avoid spending too much of their limited lifespan in treatment. In patients expected to have longer survival times, outcomes can be improved with doses > 30 Gy [8]. Moreover, the modern approaches of hippocampus-sparing and addition of memantine can significantly reduce neurocognitive decline, an important late toxicity of whole-brain-irradiation [9,10,11]. The risk of experiencing this late toxicity increases with time, and therefore, gains importance in longer-term survivors. Thus, physicians are interested in accurate knowledge of a patient’s prognosis, which can be improved with survival scores. It would be ideal to have separate scores for specific cancer types to account for differences in tumor biology and prognoses. A few tools already exist for patients with cerebral metastases from colon or rectal cancers [12,13,14,15]. However, none of these scores has been created specifically for the expanding group of older patients. This study provided the first score, particularly for older patients with cerebral metastases from colorectal cancers. Moreover, the new score was compared to existing instruments, one non-diagnosis-specific and three diagnosis-specific tools, with respect to accurately identifying patients who die within 6 months and patients who survive for 6 months or longer. The diagnosis-specific scores were the diagnosis-specific graded prognostic assessment (DS-GPA) classification for gastro-intestinal cancers, the Dziggel-Score for less radiosensitive tumors, and the WBRT-30-CRC (whole-brain radiotherapy with 30 Gy in 10 fractions for cerebral metastases from colorectal cancer) score [12,13,14,15]. These tools were not specifically created for older patients. Moreover, the DS-GPA and the Dziggel-Score also include primary tumor types different from colorectal cancer. The Evers-Score was created from a cohort of elderly patients, but includes many different tumor entities [16].

## 2. Results

### 2.1. Creation of the New Score

The median survival time in the entire series was 2 months, and the 3- and 6-month survival rates were 33% and 18%, respectively. Fifty-three patients died during the period of follow-up. The cause for death was known in 33 patients. In these patients, death was always related to intra- or extra-cerebral progression of colorectal cancer. A Karnofsky performance score (KPS) of ≥70% (*p* = 0.0002) and absence of non-cerebral metastases (*p* = 0.021) were associated with increased survival on univariate analyses (Table 1). In the additional multivariate analysis, both the KPS (hazard ratio: 2.97; 95% confidence interval: 1.61–5.65; *p* = 0.0005) and non-cerebral metastases (hazard ratio: 3.03; 95% confidence interval: 1.29–8.91; *p* = 0.009) proved to be independent predictors. Therefore, both factors were incorporated in the new scoring tool. The scoring points assigned to these factors are presented in Table 2. The scores obtained for individual patients were 0 (*n* = 26), 1 (*n* = 27), or 2 (*n* = 4) points. The corresponding survival rates were 8%, 52%, and 75%, respectively, after 3 months, and 0%, 26%, and 75%, respectively, after 6 months (Figure 1, *p* < 0.0001). Thus, the positive predictive value (PPV) to correctly identify patients dying within 6 months was 100% (0-points group), and the PPV to identify patients living for at least 6 months was 75% (2-points group). 

Of the 20 patients who had liver metastases, no patient survived longer than 5 months. Thus, the probability of dying within 6 months was 100%. In patients without liver metastases, 6-month survival rates were 0% (0 of 13 patients) in the 0-points group, 35% (7 of 20 patients) in the 1-point group, and 75% (3 of 4 patients) in the 2-points group, respectively. The PPVs were 100% and 75%, respectively, to correctly predict death within 6 months and survival for at least 6 months.

### 2.2. Comparison of the New Score to Existing Tools 

The 6-month survival probabilities of the prognostic groups of the new score, DS-GPA classification [12,13], Dziggel-Score [14], WBRT-30-CRC [15], and Evers-Score [16] are summarized in Table 3. The positive predictive values (PPVs) for the poor-prognosis groups to accurately identify patients who died within 6 months were 100% (new score), 100% (Evers-Score), 91% (WBRT-30-CRC), 87% (DS-GPA classification) and 86% (Dziggel-Score), respectively. In addition, the PPVs of the favorable-prognosis groups to accurately identify patients living for at least 6 months were 75% (new score), 75% (Dziggel-Score), 60% (WBRT-30-CRC), 45% (Evers-Score), and 33% (DS-GPA), respectively. For the DS-GPA, a range of 3.0–4.0 points was selected as the favorable-prognosis group, since only one patient had a score of >3.0. [12,13]. 

For the new score, the difference between the 0 points (poor prognosis) and the 1-point group (intermediate prognosis) was significant (*p* = 0.010, two-tailed Fisher’s exact test); a trend (*p* = 0.087) was found for the difference between the 1-point and the 2-points group (favorable prognosis). The differences between the poor-prognosis and the intermediate-prognosis group was significant for the Evers-Score (*p* = 0.006) but not for DS-GPA (*p* = 0.118), WBRT-30-CRC (*p* = 0.444), or Dziggel-Score (*p* = 1.000) groups [12,13,14,15,16]. A trend regarding the difference between the intermediate-prognosis and the favorable-prognosis group was found for the WBRT-30-CRC (p = 0.072). Differences were not significant for the Evers-Score (*p* = 0.432), DS-GPA (*p* = 1.000), and Dziggel-Score (*p* = 0.143) [12,13,14,15,16].

The new score was the only one that achieved highest PPVs (i.e., highest levels of accuracy) for both identification of patients who died within 6 months and patients who survived for at least 6 months. Moreover, the new score was the only tool that, at the same time, showed a significant difference between the poor-prognosis and the intermediate-prognosis group, and at least a trend regarding the difference between the intermediate-prognosis and the favorable-prognosis group. 

## 3. Discussion

Older patients have come more into the focus of physicians involved in cancer treatment. These patients often present with significant underlying diseases, such as diabetes, coronary heart disease, dementia, and many others. Moreover, the function of critical organs, such as the liver and kidneys, can be markedly reduced. Therefore, many of these patients cannot receive aggressive treatments like younger patients. Since the comorbidity index and organ function vary widely between individual older cancer patients, they require personalized treatment protocols to achieve the best outcome. This is particularly true for palliative situations, including metastatic spread to the brain. When physicians aim to tailor the treatment to a patient, they need to be aware of the patient’s residual lifespan. This particular knowledge can be provided by survival scores that ideally should be created for each type of cancer spreading to the brain. In the present study, we have developed the first score particularly designed for older patients with cerebral metastases from colorectal cancer. 

Two independent predictors of survival, i.e., performance status and non-cerebral metastases, formed the basis of this score, which included three prognostic groups. Patients of the 0-points group had a median survival of only 2 months, and a 3-month survival rate of only 8%. No patient reached the 6-month survival time point. Regarding this extraordinarily poor prognosis, these patients, if irradiated, should receive 20 Gy in five fractions, since this regimen lasts only five working days. According to a previous study of 442 patients treated with whole-brain irradiation alone for multiple cerebral metastases, this regimen achieved similar cerebral control and survival as the two-week 30 Gy regimen in 10 fractions [17]. Moreover, one may also consider omitting radiation therapy and use best supportive care alone for patients of the 0-points group. In a randomized trial of 538 lung cancer patients with cerebral metastases and poor survival, the addition of whole-brain irradiation (20 Gy in five fractions) for the best supportive care did not provide a significant benefit in terms of survival and quality of life [18]. The quality-adjusted life years were 46.4 days with and 41.7 days without radiation therapy. 

In the 1-point group, patients had an intermediate survival prognosis with a median survival of 4 months, and 3- and 6-month rates of 52% and 26%, respectively. No patient survived longer than 11 months. These patients may be considered for 30 Gy in 10 fractions, globally the most common regimen of whole-brain irradiation [7]. Patients of the 2-points group had much more favorable prognoses, with a median survival time of 12 months and 75% of the patients surviving for 6 months or longer. Unfortunately, this group was quite small, and accounted for only 7% of the patients in this study. These patients should be considered for whole-brain irradiation with doses >30Gy. In a previous study of 186 longer-term survivors, 40 Gy in 20 fractions, when compared to 30 Gy in 10 fractions, resulted in significantly increased 1-year cerebral control (44% vs. 28%, respectively), and survival (61% vs. 50%, respectively) [8]. A small study of 23 patients with cerebral metastases from malignant melanoma suggested that 36Gy in 12 fractions was not inferior to 40 Gy in 20 fractions regarding 6-month cerebral control (42% vs. 17%, respectively; *p* = 0.28) and survival (50% vs. 36%, respectively; *p* = 0.75) [19]. Moreover, patients of the 2-points group may be candidates for the comparably new approaches of hippocampus-sparing and the addition of memantine. Reducing the radiation dose to the hippocampi led to a reduction of cognitive decline in a phase II study of patients receiving 30 Gy in 10 fractions [9]. At 4 months after irradiation, cognitive decline was found in 7% of the 42 patients receiving hippocampus-sparing, compared to 30% in a historical control (*p* < 0.001) [9]. In a randomized trial that included 554 patients, the addition of memantine to whole-brain irradiation (37.5Gy in 15 fractions), the interval to cognitive decline was significantly longer in patients receiving the memantine (hazard ratio 0.78, *p* = 0.01) [10]. Moreover, at least at one time point with significantly better results regarding executive function, processing speed, and delayed recognition was found in the memantine group. In a recently published phase III trial of 518 patients receiving whole-brain irradiation, hippocampus-sparing and memantine were combined and compared to memantine without sparing of the hippocampi [11]. The combined approach resulted in better preservation of cognitive function without impairing cerebral progression-free survival and overall survival.

In a second step, the new score was compared to four existing tools with respect to the PPVs to correctly identify patients dying within 6 months and patients surviving for at least 6 months [12,13,14,15,16]. For identification of patients dying within 6 months, the new score achieved the best possible accuracy, i.e., a PPV of 100%. This high PPV was also achieved by the Evers-Score, which was also developed specifically for older patients [16]. These results support the idea of considering older patients a separate group and developing separate prognostic tools for them. The other three tools, which were diagnosis-specific but not designed for older patients, achieved accuracies (PPVs) of 86% (Dziggel-Score [14]), 87% (DS-GPA [12,13]), and 91% (WBRT-30-CRC [15]), which can also be considered quite high. When aiming to identify patients living for at least 6 months, all five tools were less accurate. The highest accuracy (PPV of 75%) was found for the new score and the Dziggel-Score [14], whereas the PPVs of the other three tools were even lower with 33% (DS-GPA [12,13]), 45% (Evers-Score [16]), and 60% (WBRT-30-CRC [15]), respectively. Thus, when combining both estimations, the new score appeared preferable, since it achieved the highest PPVs for both identification of patients dying within 6 months and patients surviving for at least 6 months. 

However, when using the new score or any of the existing tools, one has to be aware that they were all created in retrospective studies that might have introduced hidden selection biases. Another limitation of the present study was its comparably small sample size. Validation of the new score in a larger cohort of patients is warranted. Moreover, application of the new score appears not necessary for patients with liver metastases in addition to cerebral metastases. In the present study, the 6-month survival rate of these patients was 0%. Thus, the PPV of accurate identification of patients dying within 6 months was 100%. In patients without liver metastases, the 6-month survival rates and the PPVs of the 0-points and 2-points group were the same as in the entire cohort. Therefore, the new score will be of value for physicians who wish to tailor a treatment regimen to an elderly patient with cerebral metastases from colorectal cancer without additional liver metastases. 

## 4. Materials and Methods 

In a series of 57 older patients (aged at least 65 years [20,21]), who were treated with whole-brain irradiation for cerebral metastases from colorectal cancer between 1999 and 2019, the dose-fractionation schedule plus seven pre-radiotherapy factors were retrospectively analyzed for 3- and 6-month survival. These factors were obtained from an anonymized database and included the dose-fractionation schedule of radiation therapy (20 Gy in 5 fractions vs. 30 Gy in 10 fractions vs. 36–40 Gy in 12–20 fractions), age at irradiation of the cerebral metastases (65–73 vs. >73 years; median age was 73 years), gender, Karnofsky performance score (≤60% vs. ≥70%; median score was 60%), cancer site (colon vs. rectum), number of cerebral metastases (1 to 3 vs. at least 4 lesions; median number was 4 lesions), non-cerebral metastases (absence vs. presence), time period from diagnosis of colorectal cancer until RT of cerebral metastases (≤30 vs. >30 months; median time period was 30 months), chemotherapy before irradiation of cerebral metastases (not administered vs. administered), and control of the primary tumor (not controlled vs. controlled). The distributions of these factors are shown in Table 4. Co-morbidity was not included as an additional factor, since co-morbidity indices are usually dominated by presence of metastatic (solid) tumors, which applies to the entire cohort of this study [22,23]. In addition, co-morbidity and the Karnofsky performance score have to be considered confounding variables. The study was approved by the responsible ethics committee at the University of Luebeck. For univariate analyses, we used the Kaplan–Meier method plus the log-rank test. Significant (*p* < 0.05) factors were additionally evaluated in a multivariate analysis (Cox proportional hazard model) to considerably reduce the potential influence of confounding factors. The prognostic factors that proved to be independent were incorporated in the new survival score. For each independent prognostic factor, 0 points and 1 point, respectively, were given for worse and better survival outcomes. Finally, the points of each factor were added for each patient’s individual score. 

The new score was compared to four existing tools: three diagnosis-specific scores, including colorectal cancer patients, and one non-diagnosis-specific tool designed for older patients. Comparisons were performed regarding the positive predictive values (PPVs) to accurately identify patients who will die within 6 months and patients who will survive for at least 6 months following radiation therapy [12,13,14,15,16]. The diagnosis-specific scores included the DS-GPA classification for gastro-intestinal cancers (Table 5); the Dziggel-Score for less radiosensitive tumors, including colorectal cancers, kidney cancer, and malignant melanoma (Table 6); and the WBRT-30-CRC for colorectal cancer (Table 7). The non-diagnosis-specific tool was the Evers-Score (Table 8). Survival time was calculated from the first day of radiation therapy. The PPVs were calculated by dividing the number of true positives through the number of all patients (true positives plus false positives). Discriminations between poor-prognosis and intermediate-prognosis groups and between intermediate-prognosis and favorable-prognosis groups were investigated for each tool with the two-tailed Fisher’s exact test; *p* < 0.05 were rated significant, and *p* < 0.10 was considered to be showing a trend. 

The PPVs to correctly predict death within 6 months and survival for at least 6 months were additionally calculated in patients who had liver metastases in addition to cerebral metastases, as well as in those patients without liver metastases.

## 5. Conclusions

A new score is provided to help physicians predict the survival time of older patients with cerebral metastases form colorectal cancer and facilitate the design of a personalized treatment protocol, including dose-fractionation and technique of whole-brain irradiation. The new score was the only tool that achieved the highest accuracy among those compared for both identification of patients who died within 6 months and patients who survived for at least 6 months. Therefore, the new score can be considered preferable to the existing tools. Since the Evers-Score and the Dziggel-Score were as precise as the new score in identifying patients dying ≤6 months and surviving ≥6 months, respectively. Physicians should always consider validating the findings obtained with the new tool using these existing tools before they design an individual treatment regimen. The score appears not necessary for patients with additional liver metastases, since their 6-month survival rate was 0%.

## Figures and Tables

**Figure 1 cancers-12-00833-f001:**
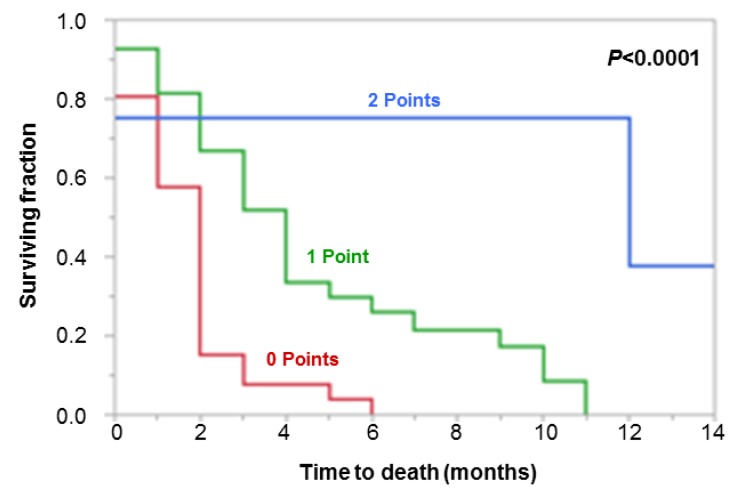
Kaplan–Meier curves of the three groups for the new score.

**Table 1 cancers-12-00833-t001:** Results of the univariate analysis of survival.

Potential Prognostic Factor	Survival at		*p*-Value
	3 Months (%)	6 Months (%)	
Radiation therapy schedule 20 Gy in 5 fractions 30 Gy in 10 fractions 36–40 Gy in 12–20 fractions	45 29 33	18 13 27	0.553
Age at irradiation of cerebral metastases 65–73 years > 73 years	37 30	20 15	0.294
Gender Female Male	33 33	13 21	0.978
Karnofsky Performance Score ≤ 60% ≥ 70%	17 52	0 37	0.0002
Cancer site Colon Rectum	28 43	14 24	0.303
Number of cerebral metastases One to three lesions At least four lesions	36 31	24 13	0.476
Non-cerebral metastases Absence Presence	75 27	38 14	0.021
Time period from diagnosis of colorectal cancer until RT of cerebral metastases ≤ 30 months > 30 months	24 43	10 25	0.605
Chemotherapy before irradiation of cerebral metastases Not administered Administered	10 38	0 21	0.076
Controlled primary tumor Not controlled Controlled	30 36	20 17	0.260

**Table 2 cancers-12-00833-t002:** Scoring points assigned to the independent prognostic factors.

Prognostic Factor	Survival at		Scoring Points
	3 Months (%)	6 Months (%)	
Karnofsky Performance Score ≤ 60% ≥ 70%	17 52	0 37	0 1
Non-cerebral metastases Absence Presence	75 27	38 14	1 0

**Table 3 cancers-12-00833-t003:** Prognosis groups of the compared tools with scoring points and 6-month survival rates [12,13,14,15,16].

Prognosis	New Score		WBRT-30-CRC		DS-GPA Scores		Dziggel-Score		Evers-Score	
	Scoring Points	6-Month Survival	Scoring Points	6-Month Survival	Scoring Points	6-Month Survival	Scoring Points	6-Month Survival	Scoring Points	6-Month Survival
Poor	0	0% (0/26)	3–4	9% (2/23)	0.0–1.0	13% (6/46)	5–8	14% (7/49)	3–6	0% (0/28)
Intermediate	1	26% (7/27)	5–9	17% (5/29)	1.5–2.5	38% (3/8)	9–1	0% (0/4)	7–9	28% (5/18)
Favorable	2	75% (3/4)	10	60% (3/5)	3.0–4.0	33% (1/3)	12–14	75% (3/4)	10–12	45% (5/11)

**Table 4 cancers-12-00833-t004:** Summary of the potential prognostic factors.

Potential Prognostic Factor	*n* Patients (%)
Radiation therapy schedule 20 Gy in 5 fractions 30 Gy in 10 fractions 36–40 Gy in 12–20 fractions	11 (19.3) 31 (54.4) 15 (26.3)
Age at RT of cerebral metastases 65–73 years > 73 years	30 (52.6) 27 (47.4)
Gender Female Male	24 (42.1) 33 (57.9)
Karnofsky Performance Score ≤ 60% ≥ 70%	30 (52.6) 27 (47.4)
Cancer site Colon Rectum	36 (63.2) 21 (36.8)
Number of cerebral metastases 1 to 3 lesions At least 4 lesions	25 (43.9) 32 (56.1)
Non-cerebral metastases Absence Presence	8 (14.0) 49 (86.0)
Time period from diagnosis of colorectal cancer until RT of cerebral metastases ≤ 30 months > 30 months	29 (50.9) 28 (49.1)
Chemotherapy before RT of cerebral metastases Not administered Administered	10 (17.5) 47 (82.5)
Controlled primary tumor Not controlled Controlled	10 (17.5) 47 (82.5)

**Table 5 cancers-12-00833-t005:** Diagnosis-specific graded prognostic assessment (DS-GPA) for colorectal cancer patients with cerebral metastases [12,13].

Prognostic Factor	GPA Scoring Criteria				
	0	1	2	3	4
Karnofsky Performance Score	<70	70	80	90	100

Prognostic groups of the DS-GPA: 0.0–1.0, 1.5–2.5, 3.0, 3.5–4.0; higher scores = better prognoses [12,13].

**Table 6 cancers-12-00833-t006:** Scoring points of the Dziggel-Score for brain lesions from less radiosensitive tumors [14].

Prognostic Factor	Scoring Points
Age <65 years ≥65 years	4 2
Karnofsky Performance Score <70 ≥70	1 4
Non-cerebral metastases Absence Presence	6 2

Prognostic groups of the Dziggel-Score: 5–8, 9–11 and 12–14 points; higher scores = better prognoses [14].

**Table 7 cancers-12-00833-t007:** Scoring points of the WBRT-30-CRC (whole-brain radiotherapy with 30 Gy in 10 fractions for cerebral metastases from colorectal cancer) Score [15].

Prognostic Factor	Scoring Points
Interval from first diagnosis of Colorectal cancer to WBRT ≤26 months >26 months	1 3
Karnofsky Performance Score ≤70% >70%	1 5
Number of brain metastases 1–3 ≥4	2 1

Prognostic groups of the WBRT-30-CRC score: 3–4, 5–6, 7–9, and 10 points; higher scores = better prognoses [15].

**Table 8 cancers-12-00833-t008:** Scoring points of the Evers-Score for elderly patients with cerebral metastases [16].

Prognostic Factor	Scoring Points
Gender Female Male	3 2
Karnofsky Performance Score <70% 70% >70%	0 4 6
Number of involved extracranial organs 0 1 ≥2	4 3 1

Prognostic groups of the Evers-Score: 3–6, 7–9, 10–12, and 13 points; higher scores = better prognoses [16].

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
