# Peer review of "An Easy-To-Use Survival Score Compared to Existing Tools for Older Patients with Cerebral Metastases from Colorectal Cancer"

_cancers, 2020, doi:10.3390/cancers12040833_

Round 1

Reviewer 1 Report

In their manuscript, the authors opt for the production of a new score for older patients with cerebreal metastasis from colorectal cancer. The aim of the score is to determine which patients will survive for more than 6 months and those who will have a poorer survival of <6 months which will aid the treating physician in determining which radiotherapy treatment modality is most suitable. 

some minor comments are found below: 

In table 4, please indicate that the age of <73 years as: 65-73 years since this score is only for older patients

in the introduction: a more thorough explanation of existing scores is highly appreciated. This could highlight the weaknesses of other scores to come up with the new score

in the methods section, could the authors please elaborate more on how they arrived at their prognostic factors? for instance, how potential confounding was controlled?

line 33 please correct "less frequently with only in 1-2%" to "less frequently in only 1-2%"

Author Response

1. Comment: In table 4, please indicate that the age of <73 years as: 65-73 years since this score is only for older patients.

Response: Age was modified accordingly in Table 4 and, additionally, in Table 1.

2. Comment: in the introduction: a more thorough explanation of existing scores is highly appreciated. This could highlight the weaknesses of other scores to come up with the new score

Response: Additional information regarding the existing tools was added to the introcuction (page 2, lines 56-63).

3. Comment: in the methods section, could the authors please elaborate more on how they arrived at their prognostic factors? for instance, how potential confounding was controlled?

Response: The M+M section has been expanded, and more information regarding the investigated factors is know provided plus the information what was done to reduce the riks of confounding variables (page 6, lines 199-215).

4. Comment: line 33 please correct "less frequently with only in 1-2%" to "less frequently in only 1-2%"

Response: The sentence has been corrected (page 1, now line 35).

Reviewer 2 Report

The manuscript by Rades et al describes a new scoring tool for survival/death for older patients with cerebral metastases from colorectal cancers. The authors compared this system to 5 existing tools and found that the described tool was better as it displayed high accuracy of patients who died within 6 months or survived for at least 6 months.

The new tool as described by the authors looks promising in evaluating treatment plan for a patient. There are some concerns which authors should address,

One major drawback of the study was small sample size and including a larger sample size could be helpful.

Moreover, as liver metastasis is more common in colorectal cancer, it would be interesting to see the accuracy of this tool under these conditions besides brain metastases.

As most of the older patients suffer from other complications (cardiometabolic, neurological etc.), results obtained from this tool could be confounded from these factors.  For example, patient survival rate would also depend on existing complications rather than solely on brain metastasis form colorectal cancer. The authors should comment on this and could also include 57 patients history of other complications if any.  

As Evers-Score and Dziggel-Scores were most precise in identifying patients dying ≤6 months and surviving ≥6 months respectively along with new tool, physicians should always consider in validating findings from new tool to existing ones before treatment plan.

Author Response

Comment: One major drawback of the study was small sample size and including a larger sample size could be helpful.

Response: This limitation of the study and the need for validation of the score in a larger cohort are now stated (page 6, lines 187-189).

Comment: Moreover, as liver metastasis is more common in colorectal cancer, it would be interesting to see the accuracy of this tool under these conditions besides brain metastases.

Response: A subgroup analysis has been added for patients who had liver metastases in addition to cerebral metastases. For these patients, the new score appears not necessary, since the 6-month survival probability of these patienst was 0%. This new finding has been added to the corresponding parts of the manuscript, namely Abstract (page 1, lines 28-29), Results (page 2, lines 81-85), Discussion (page 6, lines 189-195), M+M (page 8, lines 238-240) and Conclusions (page 8, lines 265-266).

Comment: As most of the older patients suffer from other complications (cardiometabolic, neurological etc.), results obtained from this tool could be confounded from these factors.  For example, patient survival rate would also depend on existing complications rather than solely on brain metastasis form colorectal cancer. The authors should comment on this and could also include 57 patients history of other complications if any. 

Response: The reasons why co-morbidity was not included as aseparate factor are now stated (page 6, lines 210-213). Moreover, causes of death are now mentioned (page 2, lines 68-70). 

As Evers-Score and Dziggel-Scores were most precise in identifying patients dying ≤6 months and surviving ≥6 months respectively along with new tool, physicians should always consider in validating findings from new tool to existing ones before treatment plan.

Response: This aspect was added to the Conclusions section (page 8, lines 261-265).